# Water Supply via Pedicel Reduces Postharvest Pericarp Browning of Litchi (*Litchi chinensis*) Fruit

**DOI:** 10.3390/foods13050814

**Published:** 2024-03-06

**Authors:** Fang Fang, Bin Liu, Liyu Fu, Haiyao Tang, Yanlan Li, Xuequn Pang, Zhaoqi Zhang

**Affiliations:** 1College of Horticulture, South China Agricultural University, Guangzhou 510642, China; fangfang@scau.edu.cn (F.F.); liub@scau.edu.cn (B.L.); fly940177020@163.com (L.F.); pengpengyining@163.com (H.T.); yanlanli2024@163.com (Y.L.); 2State Key Laboratory for Conservation and Utilization of Subtropical Agro-Bioresources/Guangdong Provincial Key Laboratory of Postharvest Science of Fruit and Vegetables/Engineering Research Center for Postharvest Technology of Horticultural Crops in South China, South China Agricultural University, Guangzhou 510642, China; xqpang@scau.edu.cn; 3College of Life Sciences, South China Agricultural University, Guangzhou 510642, China

**Keywords:** litchi, water supply via pedicel treatment, water loss, pericarp browning, LAC

## Abstract

Pericarp browning is the key factor for the extension of shelf life and the maintenance of the commercial value of harvested litchi fruit. Water loss is considered a leading factor of pericarp browning in litchi fruit. In this study, based on the distinct structure of litchi fruit, which is a special type of dry fruit with the aril as the edible part, the effects of water supply via pedicel (WSP) treatment on pericarp browning and the fruit quality of litchi were investigated. Compared with the packaging of the control fruit at 25 °C or 4 °C, the WSP treatment was found to significantly reduce pericarp browning and the decay of litchi fruit. The WSP-treated fruit had a higher L* value, total anthocyanin content, and pericarp water content, and the pericarp was thicker. The WSP treatment significantly suppressed the increase in the electrolyte leakage of the pericarp and maintained higher ascorbic acid (AA) contents in the aril. In addition, the WSP treatment was effective in reducing the activity and gene expression of browning-related genes *Laccase* (*ADE/LAC*) and *Peroxidase* (*POD*) during the storage period. In conclusion, the WSP treatment could be an effective method to delay pericarp browning and maintain the quality of harvested litchi fruit, and this further supports that litchi fruit has dry fruit characteristics.

## 1. Introduction

Litchi (*Litchi chinensis* Sonn.) is an attractive tropical and subtropical fruit in China and international markets for its delicious taste and high nutritional value [1]. However, litchi fruit loses its attractive color within a few days after harvest under ambient temperatures due to pericarp browning [2,3]. Pericarp browning is one of the most significant postharvest problems affecting litchi fruit; this phenomenon is responsible for reductions in fruit quality and limits shelf life, thus significantly limiting the development of the litchi industry [1,4].

Many factors related to pericarp browning have been reported, such as water loss, mechanical injuries, pathogen infections, and chilling injury, etc. Among others, water loss has long been regarded as a leading factor in inducing pericarp browning of litchi fruit [5,6,7]. We recently found that litchi pericarp has a higher water loss rate than the fruit’s other tissues, losing up to 60% of its water content after 4 days without packing [6]. The rapid loss of water in litchi pericarp is regarded to be due to the special characteristics of the fruit, which contains three parts: pericarp, aril, and seed. The only edible part of the litchi fruit is the fleshy aril, whereas most fleshy fruits have an edible pericarp layer [8]. The pericarp of litchi fruit is composed of endocarp, mesocarp, and exocarp layers, without parenchymatic fleshy tissue. Therefore, litchi fruit can be considered a dry fruit from a botanical point of view [6]. The pericarp of dry fruits is prone to water loss and becomes dry and hard soon after harvest. Similar to dry fruits, litchi pericarp loses water rapidly soon after harvesting. In addition, the pericarp and aril of the litchi fruit are separated by a transpiration barrier of the cuticle of the endocarp and connected only via the pedicel, which results in poor or no supply of water from the aril to the pericarp [8]. In addition, the proper irrigation of the litchi trees is an important factor in ensuring high-quality litchi fruit. When the litchi fruit is on the tree, the fruit is continuously supplied with water by the tree through the conducting tissues, including the pedicels. However, the pericarp of harvested litchi fruit loses water rapidly, as there is no supply of water from the plants or the aril, which triggers the pericarp browning [6,9]. Therefore, it is very important to maintain the sufficient water content of the pericarp in litchi fruit to prevent browning after harvest.

Packaging and coating are the main technologies to control water loss in fruits and vegetables postharvest. To reduce water loss from the pericarp and pericarp browning of the litchi fruit, packaging and coating materials have been used [10,11,12,13,14]. However, saturated water vapor in bags or films escapes due to the water permeability of the packaging material; thus, producers have been unable to prevent water loss in litchi fruit. In addition, the accumulation of condensed water inside the bags or films also stimulates microbial growth and decay development [7]. Coatings can also provide a moisture barrier, which helps to reduce water loss and the oxygen uptake of fruit from the environment, thus maintaining the quality of many fruits [15]. However, due to the presence of protruding semi-spherical epidermal cells and microcracks on the surface of the litchi pericarp, the effectiveness of the coating is limited. Thus, both packaging and coating technologies do not provide a satisfactory effect in preventing water loss in litchi fruit.

Litchi can be considered a dry fruit and the pericarp dehydrates when the seeds mature. In a previous study, we found that water loss plays an important role in the pericarp browning process of litchi fruit after harvest [6]. In this study, due to the pedicel being the water-conducting tissue for exogenous water supply to the pericarp, we attempt to use water supply via pedicel (WSP) treatment to reduce pericarp browning and ensure a high quality of litchi fruit by maintaining the water content in the pericarp. We also aimed to clarify the browning mechanism via the characterization of the physiological effect of the treatment and gene expression levels of pericarp browning enzymes at 25 °C or 4 °C. The WSP treatment was found to significantly reduce pericarp browning and the decay of litchi fruit. The WSP-treated litchis had a lower concentration of TSS compared with the control and maintained higher ascorbic acid (AA) contents in the aril. In addition, the WSP treatment was effective in reducing the activity and gene expression of the browning-related genes, *Laccase* (*ADE/LAC*) and *Peroxidase* (*POD*), during storage. The results help us to understand the key role of water content in pericarp browning. This study proposes the WSP treatment as a new possible method to delay pericarp browning and maintain the quality of harvested litchi fruit, a distinct type of dry fruit.

## 2. Materials and Methods

### 2.1. Plant Material and Treatments

Litchi (cv. Huaizhi) fruits were collected from an orchard located in the Conghua district of Guangzhou City, Guangdong, China, at latitude 113°53′ N, longitude 23°53′ E. In this region, litchi trees with an age of over thirty years old are planted in soils that are rich in organic matter with good drainage. The fruit development season, from April to July, is the most hot and rainy season of the year for the location with an average temperature of around 29 °C (see in website: www.tqyb.com.cn). The litchi fruits were harvested in bunches and transported to the laboratory within 2 h. The branches and leaves were removed and the fruit stalks were retained. Fruits with 80–90% maturity and with no diseases, pests, or mechanical injuries were selected. The selected fruits were treated as follows: Water supply via pedicel (WSP) treatment: the fruits were left with about 3 cm of pedicels that were inserted into 10 mL centrifuge plastic tubes with the fruit up and outside the tubes, with each tube containing 8 mL of H_2_O. The tubes were packed in boxes using a tube holder (15 fruits in each box) and sealed with 0.01 mm thick polyethylene bags. Packaging treatment (control): the fruits were placed in air-tight boxes and sealed with 0.01 mm thick polyethylene film. The samples were stored with a constant temperature of 25 °C for 8 days (room temperature, RT), or 4 °C for 28 days (low temperature, LT), with a relative humidity of 85%. Every 2 d at 25 °C and every 7 d at 4 °C for each sampling period, the samples were frozen in liquid nitrogen and placed in a refrigerator at −80 °C. Each treatment was repeated three times; a total of 30 litchis were used as a single replicate.

### 2.2. Determination of Browning Index

The browning index was determined using the method of Liu et al. [6]. A total of 30 litchi fruits were randomly selected for observation and grading. The browning index was calculated after three repetitions.

Browning index = Σ (browning class × number of fruits in each class)/total number of fruits in each treatment.

### 2.3. Color Measurement

The pericarp color was measured using a color analyzer (Konica Minolta CR-300, Tokyo, Japan). The L* values on the equatorial plane of fruit were determined. The L* values denote the darkness (0) to lightness (100) of the pericarp color. At each time point during the treatments, 3 fruits were randomly selected as 3 biological replicates, and each fruit was measured 3 times.

### 2.4. Assay of Disease Index and Marketable Fruit Rate

Decayed fruit was evaluated by the observation of visible fungal growth on the surface of the fruit. The disease index was scored on a scale of 0–4 by calculating the surface area affected, where 0 = none, 1 = slight (up to 25%), 2 = moderate (25–50%), 3 = moderately severe (50–75%), and 4 = extreme (>75%).

Disease index = Σ (disease class × number of fruits)/total number of fruits.

The marketable fruit rate (%) was calculated as fruit free of mold and rot/total fruit, expressed as a percentage.

### 2.5. Anthocyanin Content

The anthocyanin content method was performed as described by Fang et al. [16]. Briefly, 10 pericarp discs of 1 cm diameter were immersed in 5 mL of 0.1 M HCl in water. The supernatant was collected. Then, the residues were extracted twice with 5 mL of 0.1 M HCl until colorless. The 3 extractions were pooled and made up to a volume of 15 mL. The anthocyanin concentration was determined using a pH differential method [17]. A 1 mL extract was diluted in 5 mL of 0.4 M KCl-HCl buffer of pH 1 or in 5 mL of 0.4 M citric acid–Na_2_HPO_4_ buffer of pH 4.5. The absorbance of the dilutions at 510 nm was measured using a spectrophotometer (Shimadzu UV-2450, Tokyo, Japan). The content of anthocyanin (cyanidin-3-glucoside) was calculated using the following formula:Anthocyanin content (mg cm^−2^) = ΔA × L × V1 × M/ε × V2/s

L (cuvette thickness) = 1 cm; V1 (diluted extract) = 5 mL; V2 (total extract) = 15 mL; s (total square of the pericarp discs) = 7.85 cm^2^. ε = 29,600 mol^−1^ cm^−1^ and M = 445 g mol^−1^.

### 2.6. Water Content and Relative Electrolytic Leakage of Pericarp

The pericarp of litchi fruit was dried at a temperature of 65 °C until a constant weight (dry weight) was reached. The water content (%) = (fresh weight − dry weight)/fresh weight × 100.

Relative electrolyte leakage (REL) was measured according to Lin et al. [18]. A total of 20 slices of the same size from the pericarp were added to 20 mL of distilled water and kept at 25 °C for 30 min. The initial reading (Lt) was measured using a conductivity meter (HI-98304, Hanna, RI, USA). The solution was then heated at 100 °C for 20 min. The final reading (L0) was measured after cooling the boiled solution to room temperature. The REL was calculated using the formula:REL (%) = Lt/L0 × 100.

### 2.7. TSS and Ascorbic Acid Content in Aril

The total soluble solid (TSS) content of litchi fruit was determined using a digital refractometer (PR-32α, ATAGO Co., Ltd., Tokyo, Japan). The ascorbic acid (AA) content of litchi aril was determined using the 2, 6-dichlorophenol-indophenol method, and the data were expressed as g kg^−1^ [19,20].

### 2.8. Assays of Enzyme Activity

An assay of the crude enzyme activity was performed according to the method in a previous study [16]. An amount of 2.0 g of frozen litchi pericarp tissue was ground with liquid nitrogen and homogenized in 6 mL of pre-cooled 0.05 M phosphate buffer solution (pH 7.0) containing polyvinylpyrrolidone (10% sample weight) and protein inhibitors (1% *w*/*v*) (Sigma-Aldrich, St. Louis, MO, USA). The extracted mixtures were centrifuged at 9000× *g* for 20 min at 4 °C, and the supernatant was collected as the crude enzyme extract.

Laccase (LAC) activity was determined by the change in absorbance at 420 nm; the crude enzyme extract (50 μL) was added to the 2.95 mL modified universal buffer (MUB) (pH 4) containing 4 mM of 2,2′-azinobis-(-3-ethylbenzothiazoline-6-sulfononic acid) diammonium salt (ABTS). Then, the mixture was incubated for 8 min at 37 °C. The 0.01 increase in absorbance at 420 nm (A420) per minute was used as one unit (U) for enzyme activity [21].

Peroxidase (POD) activity was assayed based on the absorbance change at 470 nm owing to the production of tetraguaiacol from guaiacol with the presence of H_2_O_2_. Fresh reaction mixture was prepared by adding 700 µL of 0.05 M phosphate buffer (pH 7), 100 µL of 40 mM H_2_O_2_, and 100 µL of 20 mM guaiacol. Then, 900 µL of the mixture was reacted with 100 µL of crude enzyme extract. One unit (U) was defined as an increase in the absorbance of 0.01 at 470 nm/min [22].

### 2.9. Gene Expression Analysis by qRT-PCR

Total RNA was extracted with a Rapid RNA Extraction Kit 3.0 (Huayueyang Biotechnology Co., Ltd., Beijing, China). After checking the RNA concentration and purity, the first strand of cDNA was isolated and potential DNA contamination was removed using the PrimeScript™ RT reagent Kit with gDNA Eraser (Perfect Real Time; Takara Biomedical Technology Co., Ltd., Tokyo, Japan). The qRT-PCR was performed using SYBR Green-based PCR assay in a CFX96 Real-Time (RT) PCR system (Bio-Rad, Hercules, CA, USA). The primer sequences are listed in Table 1. The PCR conditions were 95 °C for 3 min, followed by 40 cycles of 95 °C for 15 s, 56 °C for 30 s, and 72 °C for 35 s. The relative expression of the genes was calculated according to the 2^−ΔΔCt^ method [23].

### 2.10. Statistical Analysis

Th data were expressed as the means ± standard error (SE) of the mean with three biological replicates. The differences among treatments were analyzed using the T-test and Duncan’s multiple range test at a 5% significance level (IBM SPSS Statistics 22, Armonk, NY, USA). All analyses were performed in triplicate.

## 3. Results

### 3.1. The Effects of Water Supply via Pedicel on the Appearance, Pericarp Browning, and Postharvest Quality of Litchi Fruit

To investigate the effect of water supply via pedicel (WSP) treatment on the postharvest quality of litchi fruit, the browning index, L* value, disease index, and marketable fruit rate were compared with the packaging group (control) under storage conditions of 25 °C for 8 days (room temperature, RT), or 4 °C for 28 days (low temperature, LT) (Figure 1). The WSP treatment significantly improved the appearance and storability of litchi fruit (Figure 1B,C). The increase in the pericarp browning index was effectively inhibited by the WSP treatment. The browning index of the control litchi fruit was increased rapidly to 2.8 at RT-8 d and 3.3 at LT-28 d, while the browning index of the WSP-treated litchi was maintained at 2.0 at RT-8 d and 2.2 at LT-28 d. The pericarp browning index of the litchi fruit treated with WSP was significantly lower than that of the control on days 6–8 at RT (Figure 1D) and days 21–28 at LT (Figure 1H). The color change in the pericarp during storage was studied using chromaticity L* values. Chromaticity L* values, which indicated litchi pericarp colors close to dark red or pink–red, declined continuously in the control and WSP-treated fruit during storage. The pericarp of the WSP-treated fruit had slower declines in L values than the control during the RT-8 d (Figure 1E) and LT-28 d (Figure 1I) periods, indicating that the WSP treatment maintained lightness in litchi pericarp. The disease index in the pericarp started to rapidly increase after the RT-4 d (Figure 1F) and LT-14 d (Figure 1J) periods; the WSP treatment significantly reduced the disease index during the whole storage period. The marketable fruit rate of the control fruit rapidly declined after the RT-4 d and LT-14 d periods, and the marketable fruit rate of the WSP-treated fruit was significantly higher than the control fruit (Figure 1G,K). Therefore, the WSP treatment effectively delayed pericarp browning and maintained the quality of litchi fruit.

### 3.2. The Effects of WSP Treatment on Pericarp Water Content, Thickness, Electrolytic Leakage, and Anthocyanin Content in Litchi Fruit

The pericarp water content of the control fruit gradually decreased during the RT and LT storage conditions. The water content was initially about 73% at 0 d in litchi pericarp. The water content in the pericarp of the control was reduced to 48% at RT-8 d (Figure 2A) and 60% at LT-28 d (Figure 2B), while that of the WSP-treated litchi fruit was well maintained up to 70%, indicating that the control fruits still lost quite an amount of water in the pericarp, while the WSP treatment was very effective for maintaining moisture during storage. Moreover, the pericarp thickness of the control and the WSP-treated fruit decreased during the storage. A quick decline in the pericarp thickness of the control was observed at RT-2 d (Figure 2C) and LT-7 d (Figure 2D), then showed a rapid decrease from RT-2 d to RT-8 d and a slight decrease from LT-7 d to LT-28 d. A significantly lower pericarp thickness was observed in WSP-treated litchi than in the control during whole storage at RT and LT (Figure 2C,D). The electrolytic leakage values of the pericarp of the WSP-treated and control fruit increased during storage. The REL values of the control litchi fruit exhibited a slight increase from 0 d to RT-6 d and LT-21 d, followed by a rapid increase after RT-6 d and LT-21 d, whereas the REL values in the WSP-treated pericarp were lower than that in the control pericarp during storage. The WSP treatment effectively delayed the increase in the REL values of the pericarp (Figure 2E,F). The anthocyanin contents of the control and the WSP-treated fruit decreased steadily during storage, and the anthocyanin content in the control pericarp decreased rapidly from RT-6 d and LT-14 d. The anthocyanin content of the WSP-treated fruits was always significantly higher than the control, which is consistent with the red appearance of the fruit (Figure 2G,H).

### 3.3. The Effects of WSP Treatment on TSS and Ascorbic Acid Contents in Aril of Litchi Fruit

The TSS and ascorbic acid contents are important elements in assessing the flavor and nutritional quality of litchi fruits. The contents of TSS and ascorbic acid in the aril of the litchi fruit exhibited a consistent decreasing trend during storage at 25 °C or 4 °C (Figure 3). The TSS content of the WSP-treated fruit was lower than the control during storage (Figure 3A,B), indicating that water was absorbed by the pedicel and transported to the aril. A quick decline in the ascorbic acid level in the control was observed at RT-2 d and LT-7 d, which then rapidly decreased from RT-2 d to RT-8 d, and a slight decrease was observed from LT-7 d to LT-28 d. No significant difference in the ascorbic acid content was found between the control and the WSP-treated fruits before RT-4 d and LT-7 d. Throughout the later storage period, the ascorbic acid content of the WSP-treated fruit was higher than that of the control fruit from RT-6 d to RT-8 d and LT-14 d to LT-28 d (Figure 3C,D). Therefore, the WSP treatment was effective in the maintenance of the ascorbic acid content of litchi fruit.

### 3.4. Effects of WSP Treatment on the Activity of LAC and POD Enzymes in Litchi Fruit

The activity of LAC and POD enzymes in pericarp was measured to understand the effect of the WSP treatment on pericarp browning (Figure 4). The enzyme activity of LAC peaked at RT-2 d of the control fruit, and then showed a decrease from RT-4 d to RT-8 d, while the LAC activity of the control fruit remained stable from LT-0 d to LT-21 d and increased slightly at LT-28 d. Notably, the LAC activity in the WSP-treated litchi pericarp was substantially lower than the control during the whole storage period at 25 °C or 4 °C (Figure 4A,B). The activity of the POD enzyme of the control exhibited a steady increasing trend with prolonged storage at RT, showing a rapid increase from RT-0 d to RT-4 d and a slow increase from RT-4 d to RT-8 d (Figure 4C). While the POD activity of the control remained constant from LT-0 d to LT-14 d, a rapid increase from LT-21 d to LT-28 d was observed (Figure 4D). Compared with the control, a lower level of POD activity was observed in the WPS-treated fruit during storage, and the activity of POD in the WSP-treated litchi was significantly lower than in the control from RT-4 d to RT-8 d (Figure 4C) and LT-21 d to LT-28 d (Figure 4D). Therefore, the activity of browning-related enzymes (LAC and POD) in litchi pericarp was significantly inhibited by the WSP treatment during storage.

### 3.5. WSP Treatment Reduced the Expression of Genes Related to Pericarp Browning in Litchi Fruit

During the RT or LT storage period, the expression of *LAC* (LITCHI011562) and *POD* (LITCHI011513) genes in the litchi fruit pericarp was determined (Figure 5). The expression of *LAC* in the control pericarp showed a slight increase at RT-2 d and then a rapid decrease at RT-4 d, while the expression of *LAC* remained stable in the control from LT-0 d to LT-14 d, and then showed a slight increase from LT-21 d to LT-28 d. Compared with the control at 25 °C and 4 °C, a lower expression of *LAC* was observed in the WPS treatment during storage (Figure 5A,B). The expression of *POD* in the control pericarp increased significantly after RT-6 d while remaining stable in the control throughout the storage period at LT. The expression level of *POD* of the WSP-treated fruit remained constant at lower levels than the control (Figure 5C,D). Therefore, the expression of the *LAC* and *POD* genes in the WSP-treated litchi pericarp was lower in comparison with the control.

## 4. Discussion

Pericarp browning induced by water loss is one of the major postharvest problems of litchi fruits, causing reduced market value and shelf life [1]. In this study, the WSP treatment was effective in retarding pericarp browning, maintaining fruit quality, and prolonging the storage time of litchi fruit at 25 °C and 4 °C storage conditions.

Obvious browning can be observed when there is a loss of as little as 2% water content of litchi pericarp after harvest [24]. Packaging and coating treatments have been used to reduce water loss and desiccation-induced pericarp browning [11,12,25,26]. However, the effects of the packaging and coating treatments are limited, with substantial water loss even under packed conditions [27]. The source of water loss in litchi fruit during storage is primarily from the pericarp tissue; the pericarp of harvested litchi fruit cannot absorb water from the aril because the pericarp is not connected to the aril [6]. Moreover, the semi-spherical epidermal cells may increase the surface area and increase water loss in litchi pericarp [9]. The dehydration and breakdown of the membrane were accelerated as soon as the litchi fruit was harvested from the tree because the water supply was cut off, causing microcracks on the pericarp and accelerating water loss once no protection method is applied (e.g., packaging) for water loss [28,29]. Although there are cuticular layers that cover the surface of the exocarp and endocarp of the litchi pericarp, the wax layers are discontinuous and ineffective in controlling dehydration [8]. It is known that any harvested fruit soaked in water by the pedicel will absorb water. In this study, we found that the rule is applicable to litchi fruit. By using water supply via pedicel (WSP) treatment, we observed a significant reduction in pericarp water loss in the WSP-treated fruit in comparison with the control (Figure 2A,B). The reduced water loss achieved using the WSP treatment is beneficial for maintaining the thickness and integrity of the pericarp during storage, leading to a significantly lower browning index of the WSP-treated fruit. Therefore, WSP treatment could effectively inhibit rapid water loss and maintain sufficient water content in litchi fruit pericarp, delaying pericarp browning. The structure of the pericarp is conducive to water loss. Rapid water loss induces pericarp browning and the positive effect of WSP treatment on the inhibition of pericarp browning further supports that litchi fruit has dry fruit characteristics.

Previous studies suggested that pericarp browning may be related to the oxidation of phenolics by peroxidases (PODs) and polyphenol oxidases (PPOs) in litchi fruit [1,30]. Our previous study showed that laccases are more important in pericarp browning than PPOs. Laccases could catalyze the epicatechin oxidation, an abundant polyphenol in litchi pericarp, and could catalyze the degradation of anthocyanins in the presence of EC, leading to the rapid pericarp browning and decrease in the anthocyanin content of the fruit [16,31]. As the storage progressed, the total anthocyanin content in the pericarp gradually decreased due to anthocyanin degradation, resulting in an increase in the occurrence of pericarp browning. PODs can also catalyze the oxidation of a variety of polyphenols with H_2_O_2_ in plants. In litchi, PODs are regarded to be involved in postharvest pericarp browning [2,32]. The enzymatic reactions of LACs and PODs eventually result in the formation of polymeric brown pigments [33]. The expression of LAC and POD may increase due to desiccation stress, which is the main cause of browning. In this study, the WSP treatment effectively delayed pericarp browning and anthocyanin degradation. LAC and POD enzyme activity and gene expression in the pericarp of WSP-treated litchi fruit were significantly lower than the control (Figure 4A–D), which is consistent with the preservation of the anthocyanin pigments in the fruit pericarp. Hence, the WSP treatment helps to maintain the anthocyanin concentration and prevent pericarp browning by reducing LAC and POD activity.

The biochemical and physiological changes associated with pericarp browning of litchi fruit due to water loss have been studied under packaging or coating treatments [7,34]. Cell membranes are the main sites of storage disorders, and the deterioration of the membrane reduces its ability as a diffusion barrier, resulting in the leakage of cell contents, which is reflected in a higher electrolyte leakage. The assessment of electrolyte leakage was useful in the quantification of cell damage: the higher the electrolytic leakage values, the worse the integrity of the membrane [4]. In this study, the REL of the WSP-treated fruit was lower than the control during storage at 25 °C or 4 °C (Figure 2E,F), indicating that the regional integrity of the pericarp cells was relatively complete. Studies have shown that water loss is positively associated with membrane leakage, implying that fruit desiccation causes membrane breakdown [35]. Membrane damage caused by desiccation leads to increased rates of ion leakage through microcracks, and the development of brown lesions or the appearance of watery spots on litchi fruit, which increase susceptibility to rot [36]. In this study, fruit decay was significantly lower in the WSP-treated fruit than in the control (Figure 1F–K). The significantly lower decay incidence in the WSP-treated fruit was because of the improved disease resistance through the maintenance of membrane integrity. Reduced fruit disease was closely related to fruit senescence, with improved disease resistance due to slower fruit senescence.

TSS and ascorbic acid contents are essential factors in the assessment of the flavor and nutritional quality of litchi fruit [34,37,38]. Postharvest senescence ultimately results in a reduction in TSS levels [14]. In this study, the TSS decreased steadily over the storage period. However, the WSP-treated litchi had a lower concentration of TSS compared with the control (Figure 3A,B), which may be due to the higher water content in the aril and dilution of TSS. The content of ascorbic acid was significantly higher in the WSP-treated fruit compared with the control during storage (Figure 3C,D). The significantly higher ascorbic acid content in the WSP-treated fruit may be the result of the inhibition of senescence. It is well known that ascorbic acid can be degraded under prolonged storage conditions [39]. Coating treatment reduces ascorbic acid degradation by limiting the O_2_ available for its oxidative degradation, reducing fruit senescence [13]. Therefore, the WSP treatment effectively maintained ascorbic acid content by suppressing senescence.

The WSP treatment is effective in reducing post-harvest browning of the pericarp from a scientific perspective, but it is difficult to apply for large quantities of fruit as it requires individual fruit handling in the industry. However, for the prospect of the marketing, the WSP treatment can be applied for large or high-value litchi fruit for individual packaging applications, e.g., gift-packaging. Furthermore, the positive effect of WSP treatment on pericarp browning supports that litchi fruit can be regarded as a special dry fruit with seed appendage aril as the edible part and provides a new perspective for postharvest handling and the storage of litchi fruit. Further studies are required to explore the molecular basis of how watering supply treatment affects the shelf life and quality of litchi fruit.

## 5. Conclusions

In conclusion, water supply via pedicle (WSP) treatment effectively delayed pericarp browning and maintained the quality of litchi fruit at ambient and cold temperatures, which may be related to its ability to prevent rapid water loss and maintain cell integrity. Browning-related LAC and POD enzyme activity and gene expression were inhibited and down-regulated by the WSP treatment. The WSP treatment can be applied for the single-fruit packaging of high-value litchi fruit in the market, and provides a novel perspective for postharvest handling and the storage of litchi fruit.

## Figures and Tables

**Figure 1 foods-13-00814-f001:**
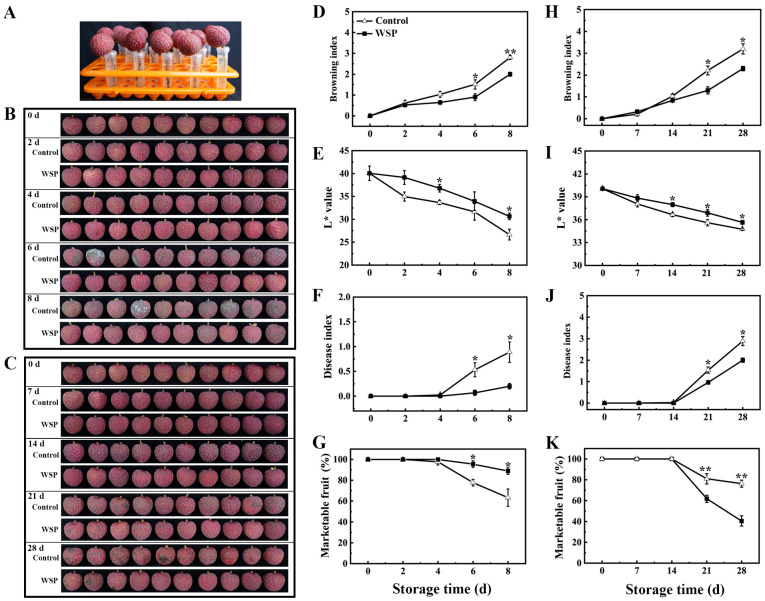
The effects of water supply via pedicel (WSP) treatment on the quality of litchi fruit during storage at 25 °C (**B**,**D**–**G**) and 4 °C (**C**,**H**–**K**). A schematic of the WSP treatment for litchi fruit (**A**). The appearance changes in litchi fruits treated with WSP stored at 25 °C (**B**) and 4 °C (**C**); pericarp browning index (**D**,**H**); chromatic L* value (**E**,**I**) for the redness of surface; disease index of pericarp (**F**,**J**); and marketable fruit rate (**G**,**K**). Each value represents the means ± SE of three replicates. Asterisks indicate that the values are significantly different between the WSP treatment and the control fruit at each time point of storage (* *p* ≤ 0.05, ** *p* ≤ 0.01).

**Figure 2 foods-13-00814-f002:**
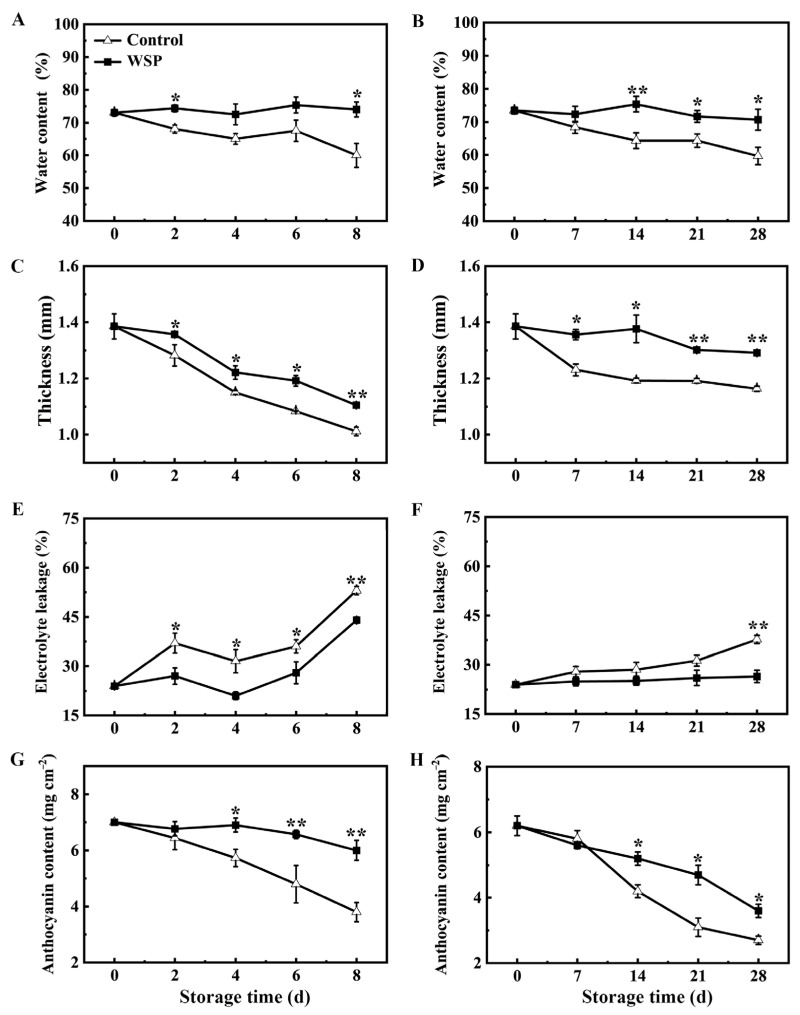
The effects of WSP treatment on the water content (**A**,**B**), thickness (**C**,**D**), electrolytic leakage (**E**,**F**), and anthocyanin content (**G**,**H**) in the pericarp of litchi fruit stored at 25 °C (**A**,**C**,**E**,**G**) and 4 °C (**B**,**D**,**F**,**H**). Statistical analyses were performed as described in Figure 1. Asterisks indicate that the values are significantly different between the WSP treatment and the control fruit at each time point of storage (* *p* ≤ 0.05, ** *p* ≤ 0.01).

**Figure 3 foods-13-00814-f003:**
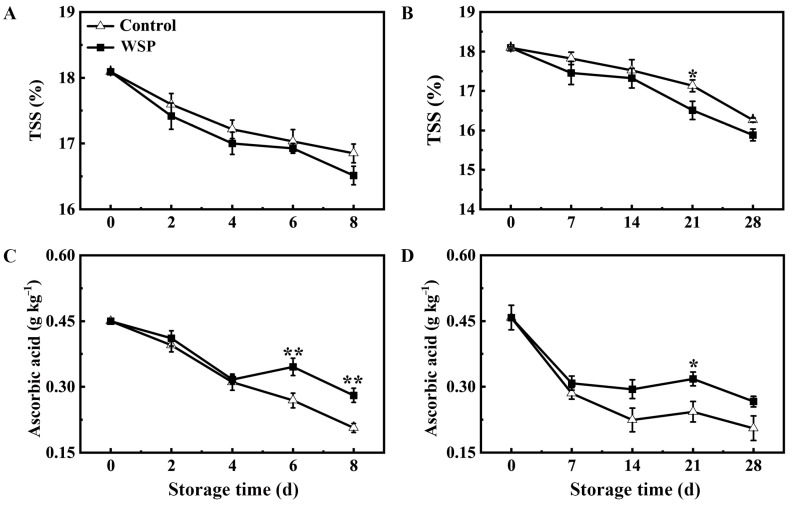
The effects of the WSP treatment on the contents of TSS (**A**,**B**) and ascorbic acid (**C**,**D**) in aril tissues of litchi fruit stored at 25 °C (**A**,**C**) and 4 °C (**B**,**D**). Statistical analyses were performed as described in Figure 1. Asterisks indicate that the values are significantly different between the WSP treatment and the control fruit at each time point of storage (* *p* ≤ 0.05, ** *p* ≤ 0.01).

**Figure 4 foods-13-00814-f004:**
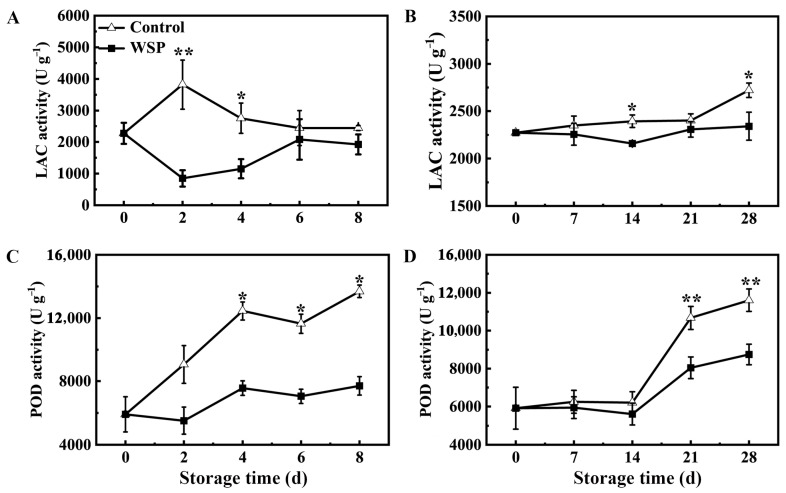
The effects of the WSP treatment on the enzyme activity of LAC (**A**,**B**) and POD (**C**,**D**) in the pericarp tissue of litchi fruit in 25 °C (**A**,**C**) and 4 °C (**B**,**D**) storage. Statistical analyses were performed as described in Figure 1. Asterisks indicate that the values are significantly different between the WSP treatment and the control fruit at each time point of storage (* *p* ≤ 0.05, ** *p* ≤ 0.01).

**Figure 5 foods-13-00814-f005:**
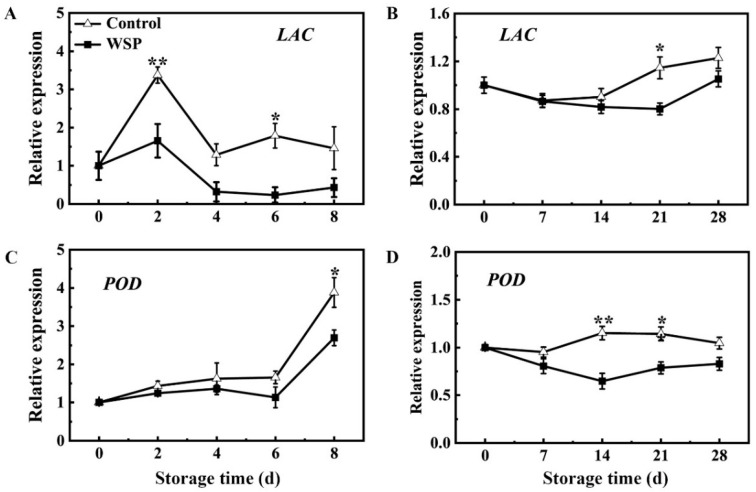
The effects of the WSP treatment on the expression of *LAC* (**A**,**B**) and *POD* (**C**,**D**) genes in litchi fruits during storage at 25 °C (**A**,**C**) and 4 °C (**B**,**D**). Statistical analyses were performed as described in Figure 1. Asterisks indicate that the values are significantly different between the WSP treatment and the control fruit at each time point of storage (* *p* ≤ 0.05, ** *p* ≤ 0.01).

**Table 1 foods-13-00814-t001:** Gene IDs and primer sequences used for RT-qPCR analysis.

Gene ID	Primer Name	Primer Sequence (5′-3′)
LITCHI007623	*Actin*-F	ACCGTATGAGCAAGGAAATCACTG
	*Actin*-R	TCGTCGTACTCACCCTTTGAAATC
LITCHI011562	*LAC*-F	ACCATTCGGCTTCATACGAC
	*LAC*-R	CAACTTATCGCCACCCAGAT
LITCHI011513	*POD*-F	GCATCAGTCACTGTACCTCTTT
	*POD*-R	TCAATGGTAGGGCAGCTTTC

## Data Availability

The original contributions presented in the study are included in the article, further inquiries can be directed to the corresponding author.

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
