# Peer review of "Water Supply via Pedicel Reduces Postharvest Pericarp Browning of Litchi (Litchi chinensis) Fruit"

_foods, 2024, doi:10.3390/foods13050814_

Round 1

Reviewer 1 Report

Comments and Suggestions for Authors

The paper by Fang et al tests the proposition that continuous water supply to the fruit prevents browning. This is shown by dipping the pedicel in water through storage. There is no doubt that this is true and it is well known that reducing water loss mitigates browning but the high humidity is conducive to disease development. By dipping the pedicel in water the fruit surface remains dry but there is hardly any desiccation of the pericarp. Such a solution is excellent from a scientific perspective but is unlikely to provide a solution for large fruit quantities at reasonable prices as it requires individual fruit handling and development of novel packaging methodologies.  

The manuscript is of good quality but requires some explanations at the text level and corrections to the text. There are suggestions for correction of the English style up to the results but further correction is required throughout.

Specific comments in the order of appearance

L4- remove and

L22 and the pericarp was thicker

L48 – claim can be arguable

L52- and such a structure

L54 – is

L63 permeability of the packaging material

L70 – could not if it is referenced, alternatively use does not seem to provide

L71 – repeating statement – rephrase

L81 – a distinct type of .. foods-2873591

L85 – it is implied as fruit were harvested in bunches - specify

L87- Fruit with no…

L88 – separate the sentences of fruit selection and treatment

L92 – specify the polymer type

L93 – what is the preservative film

L95 – which samples

L97 – the experimental setup is not very clear – specify how many fruit were used altogether and how many were assayed at each time point

L111 – delete resulting

L137 – from the pericarp were added

L149 – a previous study

L156 – was determined

L179 – the delta CT method (remove first method)

L183 – it is possible to state that all assays were done in three replicates and remove mentioning it in each section. What is not always clear is the sampling methodology – how many fruits were sampled for each method.

L195 – control litchi fruit was…

Fig. 1 – On B/C the size ruler of 1 cm is not visible and is not really necessary. On D/H Browning index or F/J disease index the Y axis can be without pericarp because it is done at the whole fruit level and can be explained the methods.  Marketable fruit can be without rate because it is simply %. The size  

Fig. 2: All Y-axis titles refer to the pericarp so for consistency remove pericarp from the axis title. No need to say relative electrolyte leakage in the axis title – specify that it is relative in the only in the methods. Spell out RT and LT in the legend. It is usually required to specify the statistical analysis in each figure separately.  

L289 – was determined

Fig. 5 – It makes more sense to change the order to C besides A and B besides D because of the similar y-axis.

Not in the results – is there information on the weight loss/gain or the water consumption by each fruit.

What were the changes in compression, was there any evidence for cracking due to the treatment.

L309 and elsewhere in the discussion. RT and LT are spelled out but are not used later. There is no need to spell out again what was spelled out before and there is no need to spell out new expressions that are used only once or twice (e.g., EC).

It is suggested that this is a method to treat litchi fruit but there is no discussion in the practicality of this method. There is no such practice in postharvest retail of fruit so at this stage the authors should reserve their claims to scientific methodology.

L319 – is it proven that the detachment of the fruit causes micro-cracks

L322 – any fruit soaked in water by the pedicel will absorb water – see papers on grapes.

L329-32 and introduction – how does the fact that water supply prevents browning proves that litchi is a "dry fruit". Litchi is a fleshy fruit with a pericarp that dries upon detachment. 70% water content at harvest is quite high and is probably not very different then orange peel water content. The structure of the pericarp is conducive to water loss.  

L333-351 – Laccase and POD are likely to be involved in browning but there expression can increase due to the desiccation stress which is known to be the primary cause for browning.

L374-76 – More likely that the difference in TSS is due to higher water content in the flesh and dilution of TSS.  

Comments on the Quality of English Language

Comments are listed above in the order of appearance. Corrections to the English were suggested for the abstract, introduction and methods but not to the results and discussion. 

Reviewer 2 Report

Comments and Suggestions for Authors

The contribution and novelty of this work to science is of medium-low interest, since it is evident and has been studied and demonstrated by many authors, that the hydration of fruits improves their general quality. From my point of view, the technique used by the authors to increase the hydration of the fruits through the pedicel is hardly viable from a practical point of view.

However, the work is well developed and confirms aspects about the quality of the litchi fruit.

The introduction is very basic. It does not indicate previous results on the parameters studied in the work and, therefore, what is new.

Examples:

Line 168: 2.9. Gene expression analysis

Line 246: TSS and ascorbic acid contents in aryl of litchi fruit

Line 264: on the activities of LAC and POD enzymes

In the introduction it would be interesting to include results, if any, of how different irrigation doses affect the quality of litchi fruits, since as the authors say in the conclusions of the work, in practice it would be the most logical way to increase the hydration of the fruits, through the xylem.

Materials and methods

The color parameter L indicates darkening. Why don't the authors show it in the figures? It is likely that some color indices such as Hue angle (hº) can indicate the browning index better than the *a parameter.

Results and discussion

It would be advisable to indicate the Tª (4ºC, or 25ºC) in the figures instead of the acronyms.

Conclusions

The conclusions are too basic and do not conclude what the contribution and novelty of the work is.

Reviewer 3 Report

Comments and Suggestions for Authors

The manuscript reflects the evident dedication and effort of the authors, and while it presents intriguing insights, there are notable areas that require improvement. Specifically, the methodology exhibits significant deficiencies, as the authors have omitted crucial details regarding sample processing for analysis. Moreover, the methods lack support from established references, and crucial factors such as cultivation conditions (soil type, climate, crop age, etc.) remain unspecified. The discussion section appears somewhat lacking, as it could benefit from a more thorough comparison and contextualization of the obtained results. Given the comprehensive methodology and applied techniques, one would anticipate a more in-depth analysis across the board.

Comments on the Quality of English Language

Moderate editing of English language required

Reviewer 4 Report

Comments and Suggestions for Authors

This study attempts to develop a new method for controlling post-harvest browning of litchi. The paper is well-structured and requires no major revisions.

1) Line 93: Please describe "thick preservative film" in detail.

2) Figure 3C, D: Ascorbic acid levels increased on day 6 of WSP. What is the reason for this?

3) Figure 4A: LAC activity drops significantly on day 2 of WSP and then increases. What is the reason for this? The trend is different from the Relative expression in Figure 5A.

Round 2

Reviewer 3 Report

Comments and Suggestions for Authors

The authors have implemented the suggested improvements. In my opinion, the article is ready for publication.